# The Influence of Propolis on Dental Plaque Reduction and the Correlation between Dental Plaque and Severity of COVID-19 Complications—A Literature Review

**DOI:** 10.3390/molecules26185516

**Published:** 2021-09-11

**Authors:** Anna Kurek-Górecka, Karolina Walczyńska-Dragon, Rafael Felitti, Aleksandra Nitecka-Buchta, Stefan Baron, Paweł Olczyk

**Affiliations:** 1Department of Community Pharmacy, Faculty of Pharmaceutical Sciences in Sosnowiec, Medical University of Silesia in Katowice, 41-200 Sosnowiec, Poland; polczyk@sum.edu.pl; 2Department of Temporomandibular Disorders, Medical University of Silesia in Katowice, 41-800 Zabrze, Poland; karolina.dragon@sum.edu.pl (K.W.-D.); aleksandra.nitecka@sum.edu.pl (A.N.-B.); sbaron@sum.edu.pl (S.B.); 3Oral Rehabilitation and Prosthodontics, Private Practice, Felix Olmedo 3716, Montevideo 11700, Uruguay; rafael.felitti@gmail.com

**Keywords:** dental plaque, dental caries, propolis, COVID-19 complications

## Abstract

Current studies suggest that cariogenic bacteria in dental plaque influence the severity of COVID-19 complications since the oral cavity is a reservoir for respiratory pathogens potentially responsible for the development of hospital-acquired pneumonia. This article focuses on the association between dental plaque and COVID-19 concerning the influence of altered oral biofilm on the risk of increased severity of SARS-CoV-2 infection. Moreover, it concentrates on the usefulness of propolis, with its apitherapeutic antibacterial properties, for treating oral bacterial infections co-occurring with SARS-CoV-2 infection. A review of the literature on PubMed, Cochrane Library and Medline between 2000 and 2021 revealed 56 published articles indicating that a link between dental plaque and COVID-19 complications was probable. Furthermore, they indicated that propolis may minimize COVID-19 severity by reducing dental plaque accumulation. The possibility that improved oral health could reduce the risk of COVID-19 complications should be of interest to scientists.

## 1. Introduction

Dental plaque is a community of microorganisms found on the surface of teeth or other hard surfaces like dentures and embedded in a matrix of polymers of both host and bacterial origin [1]. 

The microorganisms have the ability to attach to other microorganisms in a way that allows them to survive and resist host defense mechanisms or antibiotic treatments. Most of them attach to different surfaces to form some type of biofilm matrix that is highly structured and spatially organized. 

If not removed regularly, the biofilm undergoes maturation, which is connected with a progressive shift from a Gram-positive to a Gram-negative anaerobic species, which results in formation under the gingival surface, where bacteria grow profusely [2]. The pathogenic bacterial complex can lead to dental caries, periodontitis and gingivitis. Tissue injury, flossing, dental treatment and even chewing and eating can induce blood vessel injury near spaces covered in dental plaque. It allows the leakage of Gram-negative anaerobic species into the systemic bloodstream followed by bacteremia. 

In addition, dental biofilm, especially subgingival plaque in patients with periodontitis, has been associated with cardiovascular and respiratory disease and diabetes mellitus, and recently constituting a high risk for developing severe illness due to Severe Acute Respiratory Syndrome Coronavirus 2 (SARS-CoV-2) infection [3].

## 2. Materials and Methods

A literature review was performed to search for an association between dental plaque and COVID-19. Emphasizing the need for interdisciplinary cooperation during the diagnosis and treatment of patients, as well as the importance of preventive actions, we reviewed the literature in the PubMed (https://www.ncbi.nlm.nih.gov/pubmed (accessed on 19 August 2021)), Cochrane Library (http://cochranelibrary-wiley.com/cochranelibrary/search/advanced (accessed on 19 August 2021)) and Medline databases from 18 January 2020 until 30 June 2021. Trials reported in English and Polish were included. 

We selected literature on dental plaque and its role in periodontitis as well as respiratory infections including COVID-19. The following keywords were used: “dental plaque OR periodontitis OR COVID-19” and “dental plaque OR propolis OR COVID-19”. During the formulation of the search strategy we asked five questions: Is there any association between COVID-19 and oral health?Could there be a connection between dental plaque-related periodontal disease and the severity of SARS-Co-V-2 infection?Is there any junction among oral health, cytokine storm and COVID-19 complications?What is the effect of propolis on dental plaque reduction?What is the impact of propolis on the prevention of SARS-Co-V-2 infection?

After a preselection review of the titles and abstracts, those that did not meet the inclusion criteria were rejected. These criteria were as follows: the review had to include human populations and animal trials, a systematic review with or without meta-analysis, an examination of the influence of the altered oral biofilm and cytokine storm on COVID-19 complications, and the role of propolis as an antibacterial factor in oral health management. After applying the inclusion and exclusion criteria and analyzing the abstracts, two researchers conducted an independent, in-depth analysis of the remaining 56 articles. 

The quality of the studies was assessed based on the adequacy of the study design to the research objective, risk of bias, reliability of results, statistical work, and quality of reporting.

## 3. Mechanism and Risk Factors of Biofilm Formation

Dental plaque forms in an ordered way. The distinguishable stages are acquired pellicle formation; reversible adhesion, which can lead to stronger adhesin–receptor mediated attachment when bacteria loosely binds to the pellicle; co-adhesion; multiplication; and biofilm formation. As the biofilm matures it develops a primitive circulatory system that has a diverse microbial composition. Under normal conditions, it remains relatively stable in a state of “microbial homeostasis”, which does not lead to any visible disease symptoms. 

The microflora distribution is related to different conditions in different parts of the oral cavity, and this variation results, for example, from the availability of oxygen or nutrients [4]. 

Clinical studies have shown that both in patients suffering from caries or periodontitis, as well as in healthy individuals, a specific microflora occurs. The most common Gram-negative bacterial genera in the oral cavity include: *Treponema*, *Bacteroides*, *Porphyromonas*, *Prevotella*, *Capnocytophaga*, *Peptostreptococcus*, *Fusobacterium*, *Actinobacillus*, and *Eikenella*, but the proportion of these species in people with periodontal disease or caries is increased.

This sequential deposition begins with the initial colonizers (*Streptococci* and *Actinomycetes*) adhering to glycoproteins, mucins, and other proteins covering the tooth surface of the host. After brushing, these saliva proteins are deposited on the clean surface of the tooth within minutes and are called “acquired pellicle” [5]. The continued development of the biofilm depends on co-aggregation and cohesion between bacteria. Lectin-like receptors have been observed to be involved in clotting among *Streptococci*, and lipoproteins play a role in cell–cell interactions, among other early colonizers.

One of the most important *Streptococci* virulence factors is glucosyltransferases (GTFs): E.C. 2.4.1.5; sucrose: 1,6-a-d-glucan 6-a-d-glucosyltransferase; dextransucrase, because they synthesize water-soluble and insoluble glucans from sucrose. They act like an invertase when hydrolyzing sucrose into glucose and fructose and as a transferase when transferring the released glucose into the growing glucan chains [6]. 

A high level of glucosyltransferase production can also occur in an environment devoid of sucrose. The activity and properties of glucosyltransferases differ depending on whether they are free or bound to the substrate. This is of great importance in the oral cavity because GTFs exist in free form in human saliva or are incorporated into the bacterial membrane formed on the surface of the teeth [7]. 

However, only when late colonizers such as *Veillonellae, Prevotellae*, *Propionibacteria*, and some *Streptococci* begin to colonize the tooth surface, does the biofilm develop profusely. A mature biofilm consists of approximately 5–25% bacterial cells and 95–75% of a glycocalyx matrix. The oral bacterial microflora includes approximately 700 bacteria species [8]. 

Bacteria in the plaque destroy periodontal tissue directly and indirectly. The direct effect is associated with the bacterial release of proteolytic enzymes such as hyaluronidase, collagenase and elastase and toxins, as well as through the action of bacterial metabolic products. Bacteria and their products in the biofilm activate immunocompetent cells to produce and release inflammatory mediators that lead to the destruction of periodontal tissue and initiate inflammation [9]. 

Moreover, during the inflammatory phase of periodontitis, inflammatory mediators, metalloproteinases, prostaglandins, eicosanoids, kinins, cytokines and complementary activation products are released into the blood. As a result, the plasma shows increased concentrations of C-reactive protein (CRP), plasminogen activator inhibitor-1 (PAI-1), fibrinogen, tumor necrosis factor (TNF-α), interleukin-6 (IL-6), platelet-activating factor (PAF), leukocytes and platelets [10]. 

A pocket epithelium breach may open a gate for viral infection. It is known that it can result from the host immune system’s release of proinflammatory cytokines such as C5a, C3a, CL8, CXCL10, which stimulate macrophages, granulocytes, and natural killer cells, which release IL-1β, IL-6, IL-8, TNF-α, IFN-γ into the blood [9,10].

## 4. The role of Dental Plaque in The Etiology of Dental Caries and Periodontitis

It was shown that 40–50% of plaque remains after tooth brushing, and the effects of biofilm retention are clearly evident in the prevalence of caries, gingivitis and periodontal disease. Sucrose fermentation causes pH to drop rapidly to 5.0 or less at the point where the plaque contacts the tooth enamel. The enzyme invertase splits sucrose into its component glucose and fructose molecules, which are then converted to lactic acid by the glycolytic pathway [7]. 

When the pH of the dental plaque drops below 5.0–5.2, the saliva buffer cannot compensate. Tooth mineral is solubilized, thereby buffering the plaque and maintaining an environment suitable for *Streptococcus mutans* growth. Then the lactic acid that comes from the bacterial fermentation of sucrose and other carbohydrates in the diet reach the enamel surface, which begins to dissolve, releasing Ca and PO_4_ ions from the surfaces under the enamel [8].

Presence of plaque rich in bacteria (especially *Streptococcus mutans*) directly destroys the enamel layer by dissolving tooth minerals (mainly hydroxyapatite, Ca_10_(PO_4_)_6_(OH)_2_) resulting in caries. Periodontal diseases include two main conditions: gingivitis and periodontitis. Periodontitis is a complex, chronic inflammatory disease caused by an abnormal host response to bacteria from dental plaque. It affects the supporting structures of the teeth (root cementum, periodontal ligament and alveolar bone), causing irreversible attachment and bone loss, which are observed histologically and clinically. It leads finally to tooth loss. It has been estimated that periodontitis affects about 40% of the adult population. The transition from a healthy periodontium to a diseased one depends on the microbial dysbiosis and the abnormal immune host response. While the more severe forms of periodontal disease associated with alveolar bone loss are less frequent, gingivitis is widespread at all ages and is the most common form of periodontal disease. 

It was shown, that in patients with severe periodontitis, there was a significant reduction in serum albumin concentration and an increase in loss of attachment [11]. 

## 5. Possible Influence of Dental Plaque on Severe COVID-19 Complications

In late 2019, a new coronavirus SARS-CoV-2 was detected in Wuhan, China, that spread all over the world to cause a pandemic. The disease caused by SARS-CoV-2 infection was named Coronavirus Disease 2019 (COVID-19). Patients suffering from diabetes, hypertension, obesity, asthma, and kidney and liver disease are in the high-risk group for severe COVID-19 symptoms [12]. Dental plaque, which leads to oral diseases, may be a risk factor causing a severe course of COVID-19 similar to those in above mentioned chronic diseases. The argument for this hypothesis is based on data from the most recent literature. Recent studies underline the association between poor oral health and the severity of COVID-19 via direct and indirect mechanisms. The direct mechanism is connected with angiotensin-converting enzyme II (ACE-2) receptors, while the indirect mechanism is connected with inflammatory pathways and bacterial superinfections. 

ACE-2 receptors play a key role in SARS-CoV-2 entry into host cells. Periodontal disease, which is caused by a long-term accumulation of a dental plaque biofilm, exhibits a relationship with COVID-19 infection mediated by ACE-2 receptors. An association between ACE-2 and SARS-CoV-2 spike protein has been found, so patients with greater ACE-2 expression may be more susceptible to SARS-CoV-2 infection. ACE-2 receptors are present not only in blood vessels, kidney and brain but also in the tongue, buccal mucosa and gingiva. Therefore, ACE-2 in the oral mucosa may have an impact on COVID-19 infection [13,14]. Takahashi et al. confirmed that bacteria and their pathogenic factors, such as endotoxins, increases the expression of ACE-2 receptors in the altered oral microbiome [15,16]. 

Hence, the greater expression of ACE-2 receptors promotes SARS-CoV-2 infection, which Gupta et al. discovered in gingival cervicular fluid [17]. It is possible that oral diseases caused by the long-term accumulation of bacteria in dental plaque may be a risk factor for SARS-CoV-2 infection transmitted by viral droplets spread while coughing or sneezing.

The indirect mechanism, which explains the link between dental plaque and severe COVID-19 complications, is connected with inflammatory pathways due to the overproduction of proinflammatory cytokines. The increased number of cells producing cytokines and chemokines appears in the gingiva, which leads to increased levels of proinflammatory cytokines in the patient’s serum [18]. 

In 2020, COVID-19 patients presenting with oral pain, desquamative gingivitis, ulcers and blisters were reported [19]; however, mild cases of COVID-19 may not lead to oral symptoms. Severe periodontitis may worsen the progression of COVID-19 because of cytokine infiltration from periodontally diseased tissues to saliva followed by cytokine aspiration into the lungs and subsequent inflammation. Cytokines may alter the respiratory epithelium and lead to infection by respiratory pathogens. Moreover, the increased serum levels of proinflammatory cytokines—IL1-β, IL1RA, IL2, IL7, IL8, IL9, IL10, MCP1, MIP1α, FGF2, GCSF, GMCSF, IFNγ, IP10 and TNF-α—influence the severity of diseases including COVID-19 [19]. When the virus enters host cells it activates an immune response causing cytokine release syndrome or a “cytokine storm”. Thus, hypercytokinemia constitutes a consequence of SARS-CoV-2 infection and is responsible for complications such as lung injury, hypercoagulation, multiorgan failure and shock. 

Hojyo et al. demonstrated that a high severity of COVID-19 is dependent on a cytokine storm [20]. IL-6 plays a key role in the cytokines release syndrome and a high blood level is highly correlated with the COVID-19 mortality [21]. Because the production of IL-6 and IL-17 is overexpressed in periodontitis, dental biofilm, which leads to periodontal diseases, has been recognized as a cause of a cytokine storm (Figure 1). Thus, we would like to highlight the association between altered biofilm and an increased risk of complications among patients with COVID-19. 

The inflammatory pathways involved in diabetes, hypertension, obesity or cardiovascular disease are the same as those observed in periodontal diseases. It is suggested that there is a strong association between the main comorbidities and the increased risk of complications and death from COVID-19, and it may be connected with altered oral biofilm and periodontal disease. Oral diseases could be an important risk factor for COVID-19 by transmitting infectivity by droplets while breathing, talking or coughing. The oral cavity constitutes a reservoir for respiratory pathogens, therefore patients with dental plaque or periodontitis may be more likely to develop severe pneumonia [22]. It is worth noting that oral flora may affect any other respiratory infection or asthma complications. This is a consequence of the aspiration of microorganisms associated with oral diseases that lead to lower respiratory tract infection. Microbial immigration between the oral cavity and lungs increases the risk of respiratory infection and causes post-viral bacterial complications. 

Periodontal inflammation is shown by the elevated level of acute phase proteins (APPs), especially the C-reactive protein (CRP). One of proinflammatory proteins is Galectin-3, which takes part in T-cell mediated inflammation. It belongs to proteins from the group of lectins that bind to β-galactoside. Galectin-3 (Gal-3) takes part in the homeostasis of immune cells and inflammatory response. Gal-3 and CRP have been found among different predictors of COVID-19 [23,24]. In the case of Gal-3, its elevated level is connected with periodontitis [25], and this has been highlighted as a risk factor for a severe course of SARS-CoV-2 infection. 

The relationship between COVID-19 infection and Gal-3 has been observed. Inhibition of Gal-3 decreases the viral activity because it exhibits a structure that is similar to an important area in the spike protein of SARS-CoV-2: increasing the immune response and viral attachment to host cells [25,26,27,28]. 

Moreover, during persistent inflammation accompanying SARS-CoV-2 infection, the coagulation cascade increases plasma levels of D-dimers, which are produced during fibrin degradation [29]. Some of the *Streptococcus sanguis* strains located in the oral cavity, produce platelet aggregation-associated protein (PAAP) which takes part in platelet activation and aggregation. *Porphyromonas gingivalis* may release proteins similar to PAAP [30]. Therefore, the periodontal therapy in patients with initial COVID-19 symptoms may reduce the plasma D-dimer level, which is, from our point of view, very important for preventing severe COVID-19 complications [31]. 

This bacterial potential, in addition to being detrimental to periodontitis, influences distant tissues and organs.

The oral cavity also constitutes a reservoir for *Chlamydia pneumoniae*, which is an obligate intracellular Gram-negative bacterium with a biphasic developmental cycle. *C. pneumoniae* is more frequent in subgingival plaques from patients with periodontal disease than in healthy controls [32]. It can cause persistent infections, such as airway infections, asthma and atherosclerosis. Some studies have shown that patients with severe periodontitis are often carriers of *C. pneumoniae*, which may increase the risk of a host inflammatory response with serious clinical consequences.

Several mechanisms indicate the ability of oral pathogens to lead to pneumonia during SARS-CoV-2 infection. Among the mechanisms that explain the association between oral pathogens and serious COVID-19 complications are the aspiration of oral pathogens into the lower respiratory tract, modification of mucosal surfaces by salivary enzymes in the respiratory tract that facilitate colonization by pathogens, and secretion of proinflammatory cytokines during periodontitis. (Figure 2). Moreover, proinflammatory cytokines increase the adhesion of pathogens to the lung epithelium. The aspiration of pathogens during coughing may lead to the development of pneumonia requiring hospitalization [33,34]. Therefore, there are many mechanisms which, due to the action of pathogens in the oral cavity, may worsen the course of COVID-19 and lead to superinfections and further complications. It is known that the cause of patient death is not the virus itself but bacterial superinfections. It is clear that poor oral health in the form of altered biofilm is associated with an increased bacterial exchange between the lungs and mouth, which leads to post-viral complications leading to respiratory infections [35,36].

High frequency of dental plaque occurrence and periodontitis development have brought about the need to investigate their possible correlation with the severity of COVID-19 complications to introduce appropriate preventive actions [10,22]. The recent study by Yang demonstrated that periodontal treatment reduced the risk of pneumonia in patient with COVID-19 [37]. 

## 6. The Impact of Propolis on Dental Plaque Reduction

Propolis consists of a resinous substances collected by bees, and it composition depends on climate, season, location and bee species. So far, more than 300 compounds are known, but it is mainly composed of resin and balsams (50%), essential oils (10%), wax (30%) and pollen (5%) as well as other organic substances (5%). Propolis is rich in bioactive substances such as phenolic acids, flavonoids, amino acids, minerals and vitamins. The phenolic compounds include phenolic acids and esters, and flavonoids are the most numerous components with respect to quantity and type. The characteristic components of propolis in temperate regions are flavonoids (chrysin, galangin, pinocembrin, pinobanksin), minerals (magnesium, calcium, potassium, sodium, cooper, zinc, manganese and iron), and vitamins (B1, B2, B6, C and E), and phenolic compounds influence the beneficial properties of propolis on the oral cavity and teeth. Because propolis possesses bacteriostatic, bactericidal and anti-adherent properties against microorganisms, it is useful for reducing dental plaque [38,39,40,41,42,43,44].

The antibacterial activity of propolis should be considered because it stimulates the body’s immune system and is connected with direct action against microorganisms. Propolis affects the permeability of their cellular membranes by disrupting membrane potential and adenosine triphosphate (ATP) production as well as by decreasing bacterial mobility.

Moreover, the antimicrobial activity of propolis is higher for Gram-positive than Gram-negative bacteria. This phenomenon is connected with the species-specific structure of the outer membrane of Gram-negative bacteria and the production of hydrolytic enzymes by bacteria that have been inactivated by the active ingredients of propolis. 

On the other hand, one of the active propolis constituents, apigenin, was found to inhibit *Streptococcus mutans* glucosyltransferase activity. It is worth noting that another active propolis ingredient, *t*-farnesol, displays antibacterial action. The next propolis constituent, artepillin C (3,5-diprenyl-*p*-coumaric acid) is one of the numerous phenolic compounds (prenyl derivative of *p*-coumaric acid) found in Brazilian green propolis that exhibit antimicrobial properties. Artepillin C shows additional anti-inflammatory effects mediated by the modulation of NF-κB and the inhibition of prostaglandin E(2) and nitric oxide [41,42]. Moreover, numerous flavonoids in Chilean propolis, such as pinocembrin and apigenin, demonstrated higher antimicrobial activity against *Streptococcus mutans* than the polyphenol mixture or even chlorhexidine [43]. 

Furthermore, the ethanol extract of propolis demonstrated antimicrobial activity against *Streptococcus mutans*, which, together with *Lactobacillus spp.*, is the main agent responsible for biofilm formation. Dziedzic et al. showed that the antimicrobial action of the ethanol extract of propolis is based on disorganization of the cytoplasmatic membrane and cell wall, partial bacteriolysis, formation of pseudo multicellular colonies of bacteria, and inhibition of protein synthesis. According to Dziedzic et al., the minimal inhibitory concentration (MIC) for the Polish ethanol extract of propolis was estimated for *Streptococcus mutans* and was found to be equal to 1.1 mg/mL and 0.7 mg/mL for *Lactobacilli* [44]. 

Elbaz and Elsayad evaluated the antimicrobial effect of the Egyptian ethanol extract and compared it with the commercially available propolis from New Zealand. The MIC value of Egyptian propolis for *Streptococcus mutans* was 0.1–0.2 mg/mL and 0.2–0.4 mg/mL for *Lactobacilli*. The MIC value of New Zealand commercial propolis was evaluated at 0.025–0.050 mg/mL for *Streptococcus mutans* and 0.025–0.050 mg/mL for *Lactobacilli* [45]. 

Ikeno et al. showed that propolis inhibited the growth of *Streptococcus sobrinus*, *Streptococcus mutans*, and *Streptococcus cricetus* [46]. 

The research conducted by Koo et al. showed that a 3% ethanol extract of propolis decreased index plaque by about 44.7%, and researchers stated significant inhibition of insoluble polysaccharide formation of about 61.7% [47]. 

Therefore, the use of water with propolis ethanol extract as a mouthwash can reduce dental plaque. In addition, toothpaste containing propolis can prevent dental plaque formation, which was indicated by Listyasari and Santoso [48]. Table 1 summarizes the impact of active propolis constituents on dental plaque reduction.

**Table 1 molecules-26-05516-t001:** Summarization of the active propolis constituents and their impact on reduction dental plaque.

Active substances.	Action	Literature
apigenin	inhibition the activity of glucosyltransferase	[41,49]
t-farnesol, apigenin, pinocembrin, artepillin C	antibacterial action	[31,32,41]
quercetin, apigenin, pinocembrin, caffeic acid phenethyl ester (CAPE)	reduction of biofilm formation	[33]

## 7. The Impact of Propolis in Prevention against SARS-CoV-2

The antiviral activity of propolis allows it to be proposed as a potential strategy for treatment and prevention of SARS-CoV2 infection. Propolis possesses strong antiviral properties against herpesvirus, adenovirus, coronavirus, rotavirus, influenza virus and HIV. Its ingredients also show antiviral activity against COVID-19, which was confirmed in the latest research [50]. Moreover, Artepillic C, a major ingredient of Brazilian propolis, was found to be responsible for the inhibition of p21-activated kinase 1 (PAK1), a protein involved in the pathogenesis of many diseases including COVID-19 [41,51]. 

The same propolis ingredient, Artepillic C, was shown to block activation of PAK1 (RAC/CDC42-activated kinase 1), the enzyme responsible for cancer, inflammation, malaria, influenza, HIV and COVID-19. Therefore, Artepillin C acts as a PAK-1-blocker to boost the immune system to produce antibodies against the SARS-CoV-2 virus. It should be emphasized that another propolis ingredient, caffeic acid phenethyl ester (CAPE), which occurs in Egyptian or Israeli propolis, exerts the similar inhibiting PAK1 effect [52]. 

Not only is propolis a recognized PAK1-blocker, its other ingredients, such as flavonoids, possess a higher binding affinity to SARS-CoV-2, 3CL protease and SARS-CoV-2 spike protein compared to Avigan, hydroxychloroquine (HQ) and Remdesivir [53]. 

On the other hand, Refaat et al. showed that liposomes containing the Egyptian propolis extract displayed an effect similar to that of Remdesivir for inhibition of SARS-CoV-2 replication in vitro. Polyphenols from propolis were shown to inhibit viral replication and to have ability to transport Zn ions, which inhibit the viral enzymes necessary for replication in the host cells [49,54,55,56]. 

Moreover, propolis compounds such as rutin, myricetin and CAPE inhibit ACE 2 receptors, which are essential for SARS-CoV-2 virus entry [54]. Table 2 summarizes the impact of active propolis constituents on preventing SARS-CoV-2.

## 8. Conclusions

Based on recent studies and the researchers’ suggestions, hypothesis and concepts, we can underline an association between poor oral health and the severity of the course of COVID-19, via direct and indirect mechanisms. The direct mechanism is connected with angiotensin-converting enzyme II (ACE-2) receptors, while the indirect mechanism is connected with inflammatory pathways and bacterial superinfections. Among the important mechanisms that explain the association between oral pathogens and the severity of SARS-CoV-2 infection are the aspiration of oral pathogens into the lower respiratory tract and the modification of mucosal surfaces in the respiratory tract that allows the adhesion and colonization of respiratory pathogens. The pro-inflammatory cytokine secretion promotes infection by means of respiratory pathogens, and bacterial infection is one of the most frequently occurring complications. Dental biofilm, especially covering subgingival plaque, has been associated with the development of severe complications from SARS-CoV-2 infection. Dental plaque leads to an imbalance in the pathological bacteria in the mouth, contributes to bacterial superinfections, and causes complications such as pneumonia or acute respiratory distress syndrome. As presented, there may be a strong connection between dental plaque occurrence and the severity of SARS-CoV-2 infection. A reduction in dental plaque by means of propolis may diminish the risk of COVID-19 complications. Therefore, propolis is a natural substance that can bring advantages in maintaining oral health and may reduce the risk of COVID-19 complications.

## Figures and Tables

**Figure 1 molecules-26-05516-f001:**
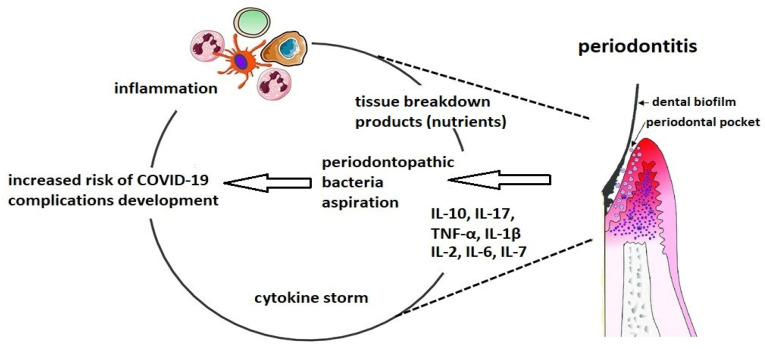
The connection between altered oral biofilm and complications of COVID-19.

**Figure 2 molecules-26-05516-f002:**
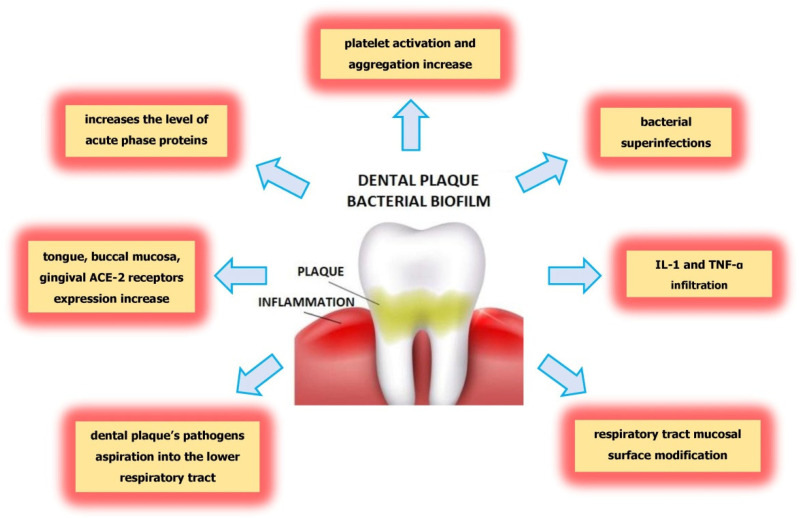
The mechanisms explaining the dental plaque influence on severity of COVID-19 complications.

**Table 2 molecules-26-05516-t002:** Summarization of the active propolis constituents and their impact on preventing against SARS-CoV-2.

Active Substances	Action	Literature
artepillin C, caffeic acid phenethyl ester (CAPE)	inhibition of PAK-1 activation	[40,41]
rutin, myricetin, caffeic acid phenethyl ester (CAPE),	inhibition of ACE2 receptors	[42]
caffeic acid, rutin, chrysin, kaemferol, galangin, fisetin, lupeol, 10-hydroxyl-2-decenoic acid	inhibition of viral replication	[38,41]

## Data Availability

Not applicable.

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
