# Peer review of "The Influence of Propolis on Dental Plaque Reduction and the Correlation between Dental Plaque and Severity of COVID-19 Complications—A Literature Review"

_molecules, 2021, doi:10.3390/molecules26185516_

Round 1
Reviewer 1 Report
The current version has a significant improvement. I agree to accept it for publication. Only one suggestion, "Covid-19" should be revised as "COVID-19".
Author Response
Respected Reviewer,
We would like to thank you for the evaluation of our resubmitted paper. According to the Reviewer’s comments we have changed Covid-19 onto COVID-19 in all article.

Reviewer 2 Report
The authors have improved their manuscript from the previously submitted version. However, the manuscript still needs to be major improvement, before being considered for publication:
- The abstract should be re-written to include: objectives, materials & methods, results and conclusions (even if this is a narrative review)
- The title should should be re-written to indicate that this is a review paper. The review in its current title misleads the readers to think it is an original research.
- Search strategy should be clearly written in a separate section titled (materials and methods) and not in the introduction. Please write a list of focused questions that helped formulate the search strategy (even though this is a narrative review, it should still follow more or less the guidelines of systematic review).
- The review includes a lot of text. Please reduce where possible.
- Figures 1 & 2 are very primitive. Besides, it is not clear how a (connection) and a (mechanism) are different. Please substitute these figures with a proper figure (an illustration and not a flow chart) that portrays the mechanisms involved. In the illustration, the periodontium .... etc. should be illustrated.
- The section (diet) seems to not fit in text. Either focus on propolis only or focus on propolis and diet that contains polyphenols. If the authors want to focus on propolis and diet, then the manuscript and title needs to be re-written accordingly.
- The authors are advised to have their manuscript revised by a native English speaker.
Author Response
Respected Rewiever
We would like to thank you for the evaluation of our resubmitted paper. We agree with the Reviewer’s comments and the text of our manuscript has been modified according to these remarks.
Once again the text has been checked and corrected by two independent a native English speakers.
We have marked the text in red colour where we made new corrections. The detailed responses are presented in attachment.

Round 2
Reviewer 2 Report
The manuscript has extensively improved from its previous versions, the new figures are detailed and clear. The efforts of the authors in addressing the comments are highly appreciated. The reviewer recommends the publication of the review in its current form.